# Constructing Semantics-Aware Adversarial Examples with a Probabilistic Perspective

**Andi Zhang**[*]
Computer Laboratory
University of Cambridge
az381@cantab.ac.uk

**Mingtian Zhang**
Centre for Artificial Intelligence
University College London
m.zhang@cs.ucl.ac.uk

**Damon Wischik**
Computer Laboratory
University of Cambridge
djw1005@cam.ac.uk

## Abstract

We propose a probabilistic perspective on adversarial examples, allowing us to embed subjective understanding of semantics as a distribution into the process of generating adversarial examples, in a principled manner. Despite significant pixel-level modifications compared to traditional adversarial attacks, our method preserves the overall semantics of the image, making the changes difficult for humans to detect. This extensive pixel-level modification enhances our method's ability to deceive classifiers designed to defend against adversarial attacks. Our empirical findings indicate that the proposed methods achieve higher success rates in circumventing adversarial defense mechanisms, while remaining difficult for human observers to detect. Code can be found at https://github.com/andiac/AdvPP.

## 1 Introduction

The purpose of generating adversarial examples is to deceive a classifier (which we refer to as victim classifier) by making minimal changes to the original data's semantic meaning. In image classification, most existing adversarial techniques ensure the preservation of adversarial example semantics by limiting their geometric distance ($\mathcal{L}_p$ distance) from the original image [39, 14, 4, 26]. While these methods can deceive classifiers using minimal geometric perturbations, they are not as successful in black-box attack scenarios. Furthermore, the recent surge in adversarial defense methods [26, 33] primarily targets geometric-based attacks, gradually reducing their effectiveness. In response to these challenges, unrestricted adversarial attacks are gaining traction as a potential solution. These methods employ more natural alterations, moving away from the small $\mathcal{L}_p$ norm perturbations typical of traditional approaches. This shift towards unrestricted modifications offers a more practical approach to adversarial attacks.

In this paper, we introduce a probabilistic perspective for adversarial examples. Through this innovative lens, both the victim classifier and geometric constraints are regarded as distinct distributions: the victim distribution and the distance distribution. Adversarial examples naturally arise as samples from the product of these distributions.

This probabilistic perspective offers an opportunity to transform traditional geometric-based constraints into semantic constraints. Traditional geometric constraints, when viewed through this

---
[*]Corresponding Author.

38th Conference on Neural Information Processing Systems (NeurIPS 2024).

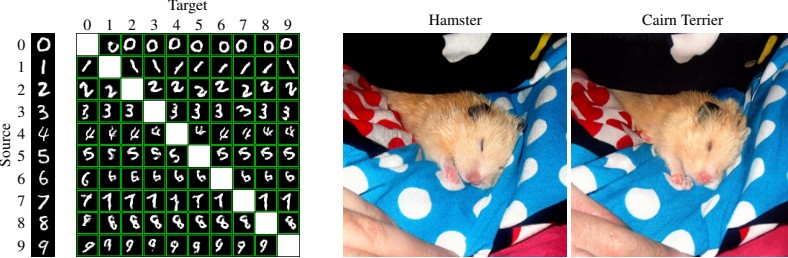

Figure 1: Adversarial examples generated by our method. Left: MNIST examples where we injected the subjective semantic understanding that scaling, translation, and distortion preserve digit meaning. The adversarial examples maintain digit interpretability while applying these transformations (see Figure 3 for comparison with other methods). Right: Adversarial example of a hamster image, leveraging semantic knowledge from pre-trained diffusion models. Despite substantial pixel modifications, the image remains natural-looking (see Figure 4 for comparison with other methods).

probabilistic lens, manifest as simple distance distributions - for instance, the squared $\mathcal{L}_2$ norm constraint naturally maps to a Gaussian distribution centered at the original image. This reveals that conventional $\mathcal{L}_2$ squared constraints implicitly define semantic similarity through a Gaussian distribution. Leveraging recent advances in probabilistic generative models (PGMs), we propose replacing this Gaussian distribution (distance distribution) with a fitted PGM. This substitution introduces a data-driven approach to defining semantic similarity, effectively transforming geometric constraints into semantic ones. To demonstrate the practical implementation of our conceptual framework, we present two approaches for constructing PGMs that capture semantic similarity:

- Our first method injects subjective semantic understanding by defining semantic-preserving transformations for the original image. We train a PGM to model the distribution of these transformed images, thereby learning the manifold of semantically equivalent variations.

- Our second method leverages the semantic knowledge embedded in pre-trained PGMs. By fine-tuning a PGM on the original image, we create a localized distribution that capture image-specific semantic variations, representing the semantic distance distribution around the original image.

These approaches can also be combined in practice. By employing appropriate PGMs as distance distributions, our method generates adversarial examples that appear more natural despite substantial geometric modifications (Figure 1). These adversarial examples demonstrate improved transferability in black-box scenarios and higher success rates against adversarial defenses.

## 2 Preliminaries

In this section, we present essential concepts related to adversarial attacks, energy-based models, and diffusion models. For detailed information on the training and sampling processes of these models, please refer to Appendix B.

### 2.1 Adversarial Examples

The notion of adversarial examples was first introduced by [39]. Let's assume we have a classifier $C : [0, 1]^n \to \mathcal{Y}$, where $n$ represents the dimension of the input space and $\mathcal{Y}$ denotes the label space. Given an image $x_{\text{ori}} \in [0, 1]^n$ and a target label $y_{\text{tar}} \in \mathcal{Y}$, the optimization problem for finding an adversarial instance $x_{\text{adv}}$ for $x_{\text{ori}}$ can be formulated as follows:

$$\min \mathcal{D}(x_{\text{ori}}, x_{\text{adv}}) \quad \text{s.t. } C(x_{\text{adv}}) = y_{\text{tar}} \text{ and } x_{\text{adv}} \in [0, 1]^n$$

Here, $\mathcal{D}$ is a distance metric employed to assess the difference between the original and perturbed images. This distance metric typically relies on geometric distance, which can be represented by $\mathcal{L}_1$, $\mathcal{L}_2$, or $\mathcal{L}_\infty$ norms.

However, solving this problem is challenging. [39] propose a relaxation of the problem: Let $\mathcal{L}(x_{\mathrm{adv}}, y_{\mathrm{tar}}) := c_1 \mathcal{D}(x_{\mathrm{ori}}, x_{\mathrm{adv}}) + c_2 f(x_{\mathrm{adv}}, y_{\mathrm{tar}})$, the optimization problem is

$$\min \mathcal{L}(x_{\mathrm{adv}}, y_{\mathrm{tar}}) \quad \text{s.t. } x_{\mathrm{adv}} \in [0, 1]^n \tag{1}$$

where $c_1$, $c_2$ are constants, and $f$ is an objective function closely tied to the classifier's prediction. For example, in [39], $f$ is the cross-entropy loss function, indicating a misclassified direction of the classifier, while [4] suggest several different choices for $f$. [39] recommend solving (1) using box-constrained L-BFGS.

## 2.2 Energy-Based Models (EBMs)

An Energy-based Model (EBM) [17, 11] involves a non-linear regression function, represented by $E_\theta$, with a parameter $\theta$. This function is known as the energy function. Given a data point, $x$, the probability density function (PDF) is given by:

$$p_\theta(x) = \frac{\exp(-E_\theta(x))}{Z_\theta} \tag{2}$$

where $Z_\theta = \int \exp(-E_\theta(x))\mathrm{d}x$ is the normalizing constant that ensures the PDF integrates to $1$.

## 2.3 Diffusion Models

Starting with data $x_0$, we define a diffusion process (also known as the forward process) using a specific variance schedule denoted by $\beta_1, \ldots, \beta_T$. This process is mathematically represented as:

$$q(x_{1:T}|x_0) = \prod_{t=1}^{T} q(x_t|x_{t-1}),$$

where each step is defined by

$$q(x_t|x_{t-1}) := \mathcal{N}(x_t; \sqrt{1 - \beta_t}\, x_{t-1}, \beta_t I)$$

where $\mathcal{N}$ is the pdf of Gaussian distributions. In this formula, the variance schedule $\beta_t \in (0, 1)$ is selected to ensure that the distribution of $x_T$ is a standard normal distribution, $q(x_T) = \mathcal{N}(x_T; 0, I)$. For convenience, we also define the notation $\alpha_t := 1 - \beta_t$ for each $t$, and $\bar{\alpha}_t := \prod_{s=1}^{t} \alpha_s$. By using the property of Gaussian distribution, we have

$$q(x_t|x_0) = \mathcal{N}(x_t; \sqrt{\bar{\alpha}_t}\, x_0, (1 - \bar{\alpha}_t)I) \tag{3}$$

The reverse process, known as the denoising process and denoted by $q(x_{t-1}|x_t)$, cannot be analytically derived. Thus, we use a parametric model, represented as $p_\theta(x_{t-1}|x_t) := \mathcal{N}(x_{t-1}; \mu_\theta(x_t, t), \Sigma_\theta(x_t, t))$, to estimate $q(x_{t-1}|x_t)$. In practice, $\mu_\theta$ and $\Sigma_\theta$ are implemented using a UNet architecture [31], which takes as input a noisy image at its corresponding timestep. For simplicity, within the rest of this paper, we will abbreviate $\mu_\theta(x_t, t)$ and $\Sigma_\theta(x_t, t)$ as $\mu_\theta(x_t)$ and $\Sigma_\theta(x_t)$ respectively.

## 3 A Probabilistic Perspective on Adversarial Examples

We propose a probabilistic perspective where adversarial examples are sampled from an adversarial distribution, denoted as $p_{\mathrm{adv}}$. This distribution can be conceptualized as a product of expert distributions [17]:

$$p_{\mathrm{adv}}(x_{\mathrm{adv}}|x_{\mathrm{ori}}, y_{\mathrm{tar}}) \propto p_{\mathrm{vic}}(x_{\mathrm{adv}}|y_{\mathrm{tar}})\, p_{\mathrm{dis}}(x_{\mathrm{adv}}|x_{\mathrm{ori}}) \tag{4}$$

where $p_{\mathrm{vic}}$ is defined as the 'victim distribution', which is based on the victim classifier and the target class $y_{\mathrm{tar}}$. $p_{\mathrm{dis}}$, on the other hand, denotes the distance distribution, where a high value of $p_{\mathrm{dis}}$ indicates a significant similarity between $x_{\mathrm{adv}}$ and $x_{\mathrm{ori}}$.

The subsequent theorem demonstrates the compatibility of our probabilistic approach with the conventional optimization problem for generating adversarial examples:

**Theorem 1.** *Given the condition that $p_{vic}(x_{adv}|y_{tar}) \propto \exp(-c_2 f(x_{adv}, y_{tar}))$ and $p_{dis}(x_{adv}|x_{ori}) \propto \exp(-c_1 \mathcal{D}(x_{ori}, x_{adv}))$, the samples drawn from $p_{adv}$ will exhibit the same distribution as the adversarial examples derived from applying the box-constrained Langevin Monte Carlo method to the optimization problem delineated in equation (1).*

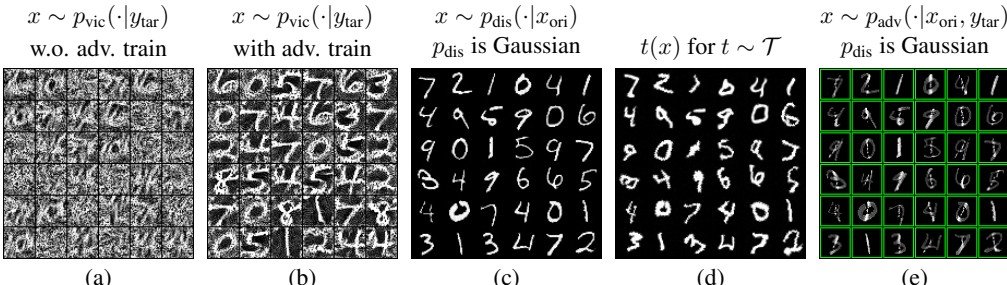

$$x \sim p_{\mathrm{vic}}(\cdot|y_{\mathrm{tar}}) \qquad x \sim p_{\mathrm{vic}}(\cdot|y_{\mathrm{tar}}) \qquad x \sim p_{\mathrm{dis}}(\cdot|x_{\mathrm{ori}}) \qquad\qquad\qquad x \sim p_{\mathrm{adv}}(\cdot|x_{\mathrm{ori}}, y_{\mathrm{tar}})$$

w.o. adv. train — with adv. train — $p_{\mathrm{dis}}$ is Gaussian — $t(x)$ for $t \sim \mathcal{T}$ — $p_{\mathrm{dis}}$ is Gaussian

(a)      (b)      (c)      (d)      (e)

Figure 2: **(a)** and **(b)** display samples drawn from $p_{\mathrm{vic}}(\cdot|y_{\mathrm{tar}})$ with the victim classifier being non-adversarially trained and adversarially trained, respectively. **(c)** showcases samples from $p_{\mathrm{dis}}(\cdot|x_{\mathrm{ori}})$ when $\mathcal{D}$ is the square of $\mathcal{L}_2$ norm. **(d)** illustrates $t(x_{\mathrm{ori}})$ for $t \sim \mathcal{T}$, where $\mathcal{T}$ represents a distribution of transformations, including TPS (see Appendix D.1), scaling, rotation, and cropping. **(e)** Samples from $p_{\mathrm{adv}}(\cdot|x_{\mathrm{ori}}, y_{\mathrm{tar}}) \propto \exp(-c_1\,\mathcal{D}(x_{\mathrm{ori}}, x_{\mathrm{adv}}))\exp(-c_2\,f(x_{\mathrm{adv}}, y_{\mathrm{tar}}))$, where $\mathcal{D}$ is the $\mathcal{L}_2$ norm, $f$ is the cross-entropy $f_{\mathrm{CE}}$, $x_{\mathrm{ori}}$ are the first 36 images from the MNIST test set, $y_{\mathrm{tar}}$ are set to 1, $c_1$ is $10^{-3}$, and $c_2$ is $10^{-2}$. A green border marks a successful attack, while red denotes failure.

The proof of the theorem can be found in Appendix A. Within the context of our discussion, we initially define $p_{\mathrm{vic}}$ and $p_{\mathrm{dis}}$ to have the same form as described in the theorem. Given this formulation, we can conveniently generate samples from $p_{\mathrm{adv}}$, $p_{\mathrm{dis}}$, and $p_{\mathrm{vic}}$ using LMC. Detailed procedures are provided in Section 5.1. As we delve further into this paper, we may explore alternative formulations for these components.

**The Victim Distribution**    $p_{\mathrm{vic}}$ is dependent on the victim classifier. As suggested by [39], $f$ could be the cross-entropy loss of the classifier. We can sample from this distribution using Langevin dynamics. Figure 2(a) presents samples drawn from $p_{\mathrm{vic}}$ when the victim classifier is subjected to standard training, exhibiting somewhat indistinct shapes of the digits. This implies that the classifier has learned the semantics of the digits to a certain degree, but not thoroughly. In contrast, Figure 2(b) displays samples drawn from $p_{\mathrm{vic}}$ when the victim classifier undergoes adversarial training. In this scenario, the shapes of the digits are clearly discernible. This observation suggests that we can obtain meaningful samples from adversarially trained classifiers, indicating that such classifiers depend more on semantics, which corresponds to the fact that an adversarially trained classifier is more difficult to attack. A similar observation concerning the generation of images from an adversarially trained classifier has been reported by [34, 55].

**The Distance Distribution**    $p_{\mathrm{dis}}$ relies on $\mathcal{D}(x_{\mathrm{ori}}, x_{\mathrm{adv}})$, representing the distance between $x_{\mathrm{adv}}$ and $x_{\mathrm{ori}}$. By its nature, samples that are closer to $x_{\mathrm{ori}}$ may yield a higher $p_{\mathrm{dis}}$, which is consistent with the objective of generating adversarial samples. For example, if $\mathcal{D}$ represents the square of the $\mathcal{L}_2$ norm, then $p_{\mathrm{dis}}$ becomes a Gaussian distribution with a mean of $x_{\mathrm{ori}}$ and a variance determined by $c_1$. Figure 2 (c) portrays samples drawn from $p_{\mathrm{dis}}$ when $\mathcal{D}$ is the square of the $\mathcal{L}_2$ distance and $c_1$ is relatively large. The samples closely resemble the original images, $x_{\mathrm{ori}}$s. This is attributed to the fact that each sample converges near the Gaussian distribution's mean, which corresponds to the $x_{\mathrm{ori}}$s.

**The Product of the Distributions**    Samples drawn from $p_{\mathrm{adv}}$ tend to be concentrated in the regions of high density resulting from the product of $p_{\mathrm{vic}}$ and $p_{\mathrm{dis}}$. As is discussed, a robust victim classifier possesses generative capabilities. This means the high-density regions of $p_{\mathrm{vic}}$ are inclined to generate images that embody the semantics of the target class. Conversely, the dense regions of $p_{\mathrm{dis}}$ tend to produce images reflecting the semantics of the original image. If these high-density regions of both $p_{\mathrm{vic}}$ and $p_{\mathrm{dis}}$ intersect, then samples from $p_{\mathrm{adv}}$ may encapsulate the semantics of both the target class and the original image. As depicted in Figure 2 (e), the generated samples exhibit traces of both the target class and the original image. From our probabilistic perspective, the tendency of the generated adversarial samples to semantically resemble the target class stems from the generative ability of the victim distribution. Therefore, it is crucial to construct appropriate $p_{\mathrm{dis}}$ and $p_{\mathrm{vic}}$ distributions to minimize any unnatural overlap between the original image and the target class. We will focus on achieving this throughout the remainder of this paper.

# 4 Semantics-aware Adversarial Examples

Under the probabilistic perspective, the distance distribution $p_{\text{dis}}$ is not necessarily based on a explicitly defined distance $\mathcal{D}$. Instead, the primary role of $p_{\text{dis}}$ is to ensure that samples generated from $p_{\text{dis}}(\cdot|x_{\text{ori}})$ closely resemble the original data point $x_{\text{ori}}$. With this concept in mind, we gain the flexibility to define $p_{\text{dis}}$ in various ways. In this work, we construct $p_{\text{dis}}$ using a probabilistic generative model (PGM). By utilizing this data-driven distance distribution and choosing a proper victim distribution, we can generate adversarial examples that exhibit more natural transformations in terms of semantics. These are referred to as semantics-aware adversarial examples. Moving forward, given that the distance $\mathcal{D}$ is implicitly learned within $p_{\text{dis}}$, we set $c_1 = 1$ for simplicity, and henceforth, use $c$ to denote $c_2$.

## 4.1 Data-driven Distance Distributions

We present two methods to develop the distribution $p_{\text{dis}}(\cdot|x_{\text{ori}})$ centered on $x_{\text{ori}}$: The first relies on a subjective understanding of semantic similarity, while the second leverages the semantic generalization capabilities of contemporary PGMs.

**Semantics-Invariant Data Augmentation**   Consider $\mathcal{T}$, a set of transformations we subjectively believe to maintain the semantics of $x_{\text{ori}}$. We train a PGM on the dataset $\{t_1(x_{\text{ori}}), t_2(x_{\text{ori}}), \dots\}$, where each $t_i$ is a sample from $\mathcal{T}$, thereby shaping the distribution $p_{\text{dis}}$. Through $\mathcal{T}$, individuals can incorporate their personal interpretation of semantics into $p_{\text{dis}}$. For instance, if one considers that scaling, rotation and TPS distortion (Appendix D.1) do not alter an image's semantics, these transformations are included in $\mathcal{T}$.

**Fine-Tuning Pretrained PGMs**   Contemporary PGMs demonstrate remarkable semantic generalization capabilities when fine-tuned on a single object or image [32, 20]. Leveraging this trait, we propose fine-tuning the PGM on the given image $x_{\text{ori}}$. The distribution of the fine-tuned model then closely aligns with $x_{\text{ori}}$, while still facilitating robust semantic generalization.

## 4.2 Victim Distributions

The victim distribution $p_{\text{vic}} \propto \exp(c\, f(x_{\text{adv}}, y_{\text{tar}}))$ is influenced by the choice of function $f$. Let $g_\phi : [0,1]^n \rightarrow \mathbb{R}^{|\mathcal{Y}|}$ be a classifier that produces logits as output with $\phi$ representing the neural network parameters, $n$ denoting the dimensions of the input, and $\mathcal{Y}$ being the set of labels (the output of $g_\phi$ are logits). [39] suggested using cross-entropy as the function $f$, which can be expressed as

$$f_{\text{CE}}(x, y_{\text{tar}}) := -g_\phi(x)[y_{\text{tar}}] + \log \sum_y \exp(g_\phi(x)[y]) = -\log \sigma(g_\phi(x))[y_{\text{tar}}]$$

where $\sigma$ denotes the softmax function.

[4] explored and compared multiple options for $f$. They found that, empirically, the most efficient choice of their proposed $f$s is:

$$f_{\text{CW}}(x, y_{\text{tar}}) := \max(\max_{y \neq y_{\text{tar}}} g_\phi(x)[y] - g_\phi(x)[y_{\text{tar}}], 0).$$

In this study, we employ $f_{\text{CW}}$ for the MNIST dataset and $f_{\text{CE}}$ for the ImageNet dataset. A detailed discussion on this choice is provided in Section 8.1.

# 5 Concrete PGM Implementations

Fundamentally, any probabilistic generative model (PGM) is capable of fitting the distance distribution $p_{\text{dis}}$. However, for efficient sampling from $p_{\text{adv}}$, which is the multiplication of $p_{\text{vic}}$ and $p_{\text{dis}}$ as introduced in (4), we recommend employing sampling techniques based on the score $s = \nabla \log p_{\text{adv}}(x_{\text{adv}}|x_{\text{ori}}, y_{\text{tar}})$. Energy-based models and diffusion models are particularly effective in providing these scores. Therefore, in this study, we utilize these models to represent $p_{\text{dis}}$.

## 5.1 Generating Adversarial Examples Using Energy-Based Models

The distance distribution $p_{\mathrm{dis}}(\cdot|x_{\mathrm{ori}})$ can be modeled using energy-based models (EBMs). For a given $x_{\mathrm{ori}}$, we train or fine-tune an EBM in the vicinity of $x_{\mathrm{ori}}$ to represent this distance distribution. Let $E_\theta$ denote the energy in the EBM. Consequently, the adversarial distribution is expressed as:

$$p_{\mathrm{adv}}(x_{\mathrm{adv}}|x_{\mathrm{ori}}, y_{\mathrm{tar}}) \propto e^{-cf(x_{\mathrm{adv}}, y_{\mathrm{tar}})} e^{-E_\theta(x_{\mathrm{adv}})}$$

and the corresponding score is:

$$\nabla_{x_{\mathrm{adv}}} \log p_{\mathrm{adv}}(x_{\mathrm{adv}}|x_{\mathrm{ori}}, y_{\mathrm{tar}}) = -c\nabla_{x_{\mathrm{adv}}} f(x_{\mathrm{adv}}, y_{\mathrm{tar}}) - \nabla_{x_{\mathrm{adv}}} E_\theta(x_{\mathrm{adv}})$$

Utilizing Langevin dynamics (Appendix B.1), we can sample from this adversarial distribution. The process is detailed in Algorithm 1 (Appendix C).

## 5.2 Generating Adversarial Examples Using Diffusion Models

To enhance generation quality and enable the creation of higher resolution images, we frame the construction of the adversarial distribution within a diffusion model context. Given an original image $x_{\mathrm{ori}}$ and a target class $y_{\mathrm{tar}}$, we employ a diffusion model $p_{\theta_{\mathrm{ori}}}(x_{t-1}|x_t) := \mathcal{N}(x_{t-1}; \mu_{\theta_{\mathrm{ori}}}(x_t), \Sigma_{\theta_{\mathrm{ori}}}(x_t))$ at time step $t-1$, where $\theta_{\mathrm{ori}}$ denotes the parameters of the model when trained or fine-tuned on $x_{\mathrm{ori}}$. This diffusion model is used to approximate $p_{\mathrm{dis}}(\cdot|x_{\mathrm{ori}})$. For simplicity, we will refer to $\theta_{\mathrm{ori}}$ as $\theta$ throughout this paper, assuming no confusion arises from this notation. Then, letting $x_0 = x_{\mathrm{adv}}$, the adversarial distribution is formulated as:

$$p_{\mathrm{adv}}(x_0|x_{\mathrm{ori}}, y_{\mathrm{tar}}) \propto p_{\mathrm{vic}}(x_0|y_{\mathrm{tar}}) p_{\mathrm{dis}}(x_0|x_{\mathrm{ori}}) = p_{\mathrm{vic}}(x_0|y_{\mathrm{tar}}) \int p(x_T) \prod_{t=1}^{T} p_\theta(x_{t-1}|x_t) dx_{1\ldots T}$$

where $p(x_T)$ is $N(0, I)$ and the victim distribution $p_{\mathrm{vic}}(x|y_{\mathrm{tar}}) \propto \exp(-cf(x, y_{\mathrm{tar}}))$. However, sampling $x_0$ in this form is challenging in the denoising order of diffusion models. Therefore, we incorporate the $p_{\mathrm{vic}}$ term within the product:

$$\int p(x_T) \prod_{t=1}^{T} p_{\mathrm{vic}}(x_0|y_{\mathrm{tar}})^{1/T} p_\theta(x_{t-1}|x_t) dx_{1\ldots T}$$

As each denoising step cannot predict $x_0$ when sampling $x_{t-1}$, we employ Tweedie's approach [12, 22, 16] to estimate $x_0$ given $x_t$:

$$\hat{x}_{0|t} = \frac{1}{\sqrt{\bar{\alpha}_t}}(x_t - \sqrt{1 - \bar{\alpha}_t}\, \epsilon_\theta(x_t)) \tag{5}$$

with $\epsilon_\theta(x_t)$ obtainable through the reparametrization trick from [18]:

$$\epsilon_\theta(x_t) = \frac{\sqrt{1 - \bar{\alpha}_t}}{\beta_t}(x_t - \sqrt{\alpha_t}\, \mu_\theta(x_t)) \tag{6}$$

This leads to a practical expression for the adversarial distribution:

$$\int p(x_T) \prod_{t=1}^{T} p_{\mathrm{vic}}(\hat{x}_{0|t}|y_{\mathrm{tar}})^{1/T} p_\theta(x_{t-1}|x_t) dx_{1\ldots T}$$

In each denoising step, we sample $x_{t-1}$ from the distribution $p_{\mathrm{vic}}(\hat{x}_{0|t}|y_{\mathrm{tar}})^{1/T} p_\theta(x_{t-1}|x_t)$. The following theorem demonstrates that this distribution approximates a Gaussian distribution:

**Theorem 2.** *Let $p_{vic}(x|y_{tar}) \propto \exp(-cf(x, y_{tar}))$ and $p_\theta(x_{t-1}|x_t) = \mathcal{N}(x_{t-1}; \mu_\theta(x_t), \Sigma_\theta(x_t))$, we have*

$$p_{vic}(\hat{x}_{0|t}|y_{tar})^{1/T} p_\theta(x_{t-1}|x_t) \approx \mathcal{N}(x_{t-1}; \mu_\theta(x_t) + \frac{c}{T}\Sigma_\theta(x_t)g, \Sigma_\theta(x_t))$$

*where $g = -\nabla_{x_{t-1}} f(\hat{x}_{0|t}, y_{tar})|_{x_{t-1}=\mu_\theta(x_t)}$ and $\hat{x}_{0|t}$ is defined in Equation (5).*

For the proof, refer to Appendix A. Building upon Theorem 2, and assuming $\nabla_{x_{t-1}} f(\hat{x}_{0|t}, y_{\mathrm{tar}})|_{x_{t-1}=\mu_\theta(x_t)} \approx \nabla_{x_{t-1}} f(\hat{x}_{0|t}, y_{\mathrm{tar}})|_{x_{t-1}=x_t}$, in line with the assumption made by [10], we introduce Algorithm 2 (Appendix C). This algorithm is designed to sample from the adversarial distribution $p_{\mathrm{adv}}$ as formulated within the context of diffusion models.

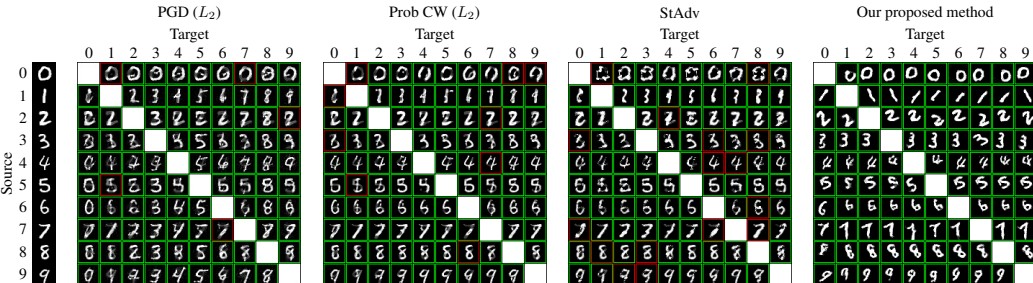

Figure 3: Comparative visual analysis of PGD, Prob CW, StAdv, and our proposed method applied to MNIST. The surrogate classifier used is MadryNet with adversarial training. Images are framed with a green border to indicate a successful white-box attack, whereas a red border signifies a failed attack.

# 6 Experiments

## 6.1 MNIST

**Setting** We use an energy-based model (EBM) to model the distance distribution $p_{\text{dis}}(\cdot|x_{\text{ori}})$ for a given original image $x_{\text{ori}}$. This EBM is specifically trained on a set of transformations of $x_{\text{ori}}$, denoted as $\{t_1(x_{\text{ori}}), t_2(x_{\text{ori}}), \dots\}$, where each $t_i$ represents a sample from the transformation distribution $\mathcal{T}$. This distribution includes a variety of transformations such as translations, rotations, and TPS (Thin Plate Spline) transformations (Appendix D.1). Examples of these transformed MNIST images are showcased in Figure 2 (d). To produce high-quality adversarial examples for MNIST, we employ rejection sampling and sample refinement techniques, as detailed in Appendix D.

For the victim distribution $p_{\text{vic}}$, we choose the adversarially trained Madrynet as our victim (surrogate) classifier. We use $f_{\text{CW}}$ to represent the function $f$ in the victim distribution, as detailed in Section 4.2.

We benchmark our method against several approaches: PGD [26], ProbCW (which employs a Gaussian distribution for $p_{\text{dis}}$ and $f_{\text{CW}}$ for $p_{\text{vic}}$), and stAdv [46].

Table 1: Success rate (%) of the methods on MNIST.

|  | PGD | ProbCW | stAdv | OURS |
|---|---|---|---|---|
| Human Anno. | 88.4 | 89.3 | 90.1 | **92.6** |
| **White-box** | | | | |
| MadryNet Adv | 25.3 | 30.2 | 29.4 | **100.0** |
| **Transferability** | | | | |
| MadryNet noAdv | 15.1 | 17.4 | 16.3 | **61.4** |
| Resnet noAdv | 10.2 | 10.9 | 12.5 | **24.3** |
| **Adv. Defence** | | | | |
| Resnet Adv | 7.2 | 8.8 | 11.5 | **18.5** |
| Certified Def | 10.7 | 12.3 | 20.8 | **39.2** |

**Quantitative result** We select 20 images from each class in the MNIST test set as the original images. For each image, we generate one adversarial example for each target class, excluding the image's true class. This yields a total of $20 \times 10 \times 9 = 1800$ adversarial examples for each method. The parameters of each method are adjusted to ensure approximately 90% of the adversarial examples accurately reflect the original concept of $x_{\text{ori}}$. The effectiveness of our adversarial examples is evaluated against the adversarially trained Madrynet under white-box conditions, with results displayed in the 'MadryNet Adv' row of Table 1. Additionally, we task 5 human annotators with classifying these adversarial examples, considering an example to be successfully deceptive if the annotator identifies its original class. The annotators' success rates are shown in the 'Human Anno.' row of Table 1. We also assess the transferability and the success rate of the examples against defensive methods, with these outcomes detailed in Table 1. Notably, the term 'Certified Def' denotes the defense method introduced by [45].

Table 1 demonstrates that our proposed method achieves a higher success rate in white-box scenarios and greater transferability to other classifiers and defense methods, all while preserving the meaning of the original image. The white-box success rate of our method reaches 100% due to the implementation of rejection sampling, as introduced in Appendix D.2.

**Qualitative result** Unlike the quantitative experiment, here we adjust the parameters so that the vast majority of examples can just barely deceive the classifier. The adversarial examples thus generated

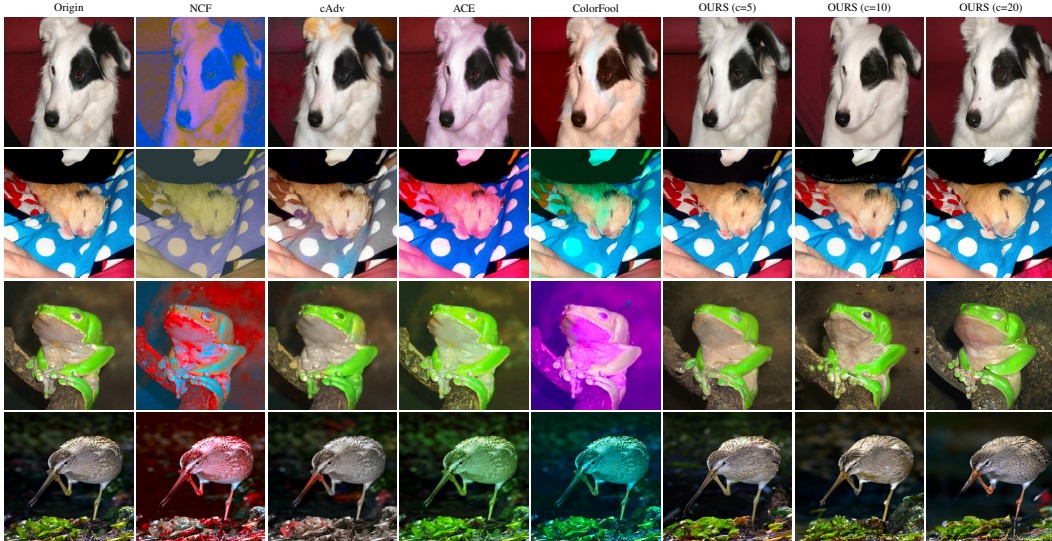

Figure 4: Comparative visual analysis of NCF, cAdv, ACE, ColorFool and our proposed method applied to Imagenet. The surrogate classifier used is ResNet50. For additional examples, refer to Appendix I.

Table 2: Success rate (%) of the methods on Imagenet.

|  | NCF | cAdv | ACE | ColorFool | OURS ($c = 5$) | OURS ($c = 10$) | OURS ($c = 20$) |
|---|---|---|---|---|---|---|---|
| Human Anno. | 2.6 | 16.9 | 11.2 | 6.7 | **31.2** | 28.3 | 25.4 |
| **White-box** | | | | | | | |
| Resnet 50 | **94.2** | 93.3 | 91.2 | 90.4 | 84.5 | 88.4 | 91.3 |
| **Transferability** | | | | | | | |
| VGG19 | **83.7** | 71.0 | 73.5 | 72.8 | 70.8 | 74.4 | 79.7 |
| ResNet 152 | **71.8** | 61.1 | 55.4 | 54.9 | 57.3 | 64.2 | 67.0 |
| DenseNet 161 | **63.9** | 54.0 | 45.1 | 41.3 | 45.0 | 52.4 | 55.3 |
| Inception V3 | **72.4** | 60.1 | 57.5 | 57.4 | 58.1 | 64.1 | 66.9 |
| EfficientNet B7 | **72.9** | 58.0 | 56.3 | 62.6 | 60.4 | 61.6 | 66.0 |
| **Adversarial Defence** | | | | | | | |
| Inception V3 Adv | **61.1** | 48.9 | 40.3 | 41.9 | 43.4 | 47.2 | 51.4 |
| EfficientNet B7 Adv | **50.3** | 42.9 | 34.7 | 36.1 | 37.7 | 40.4 | 44.2 |
| Ensemble IncRes V2 | **53.3** | 45.2 | 36.6 | 35.6 | 39.7 | 42.7 | 46.5 |
| Average | **66.2** | 55.2 | 49.9 | 50.3 | 51.6 | 55.9 | 59.6 |

are displayed in Figure 3. From this figure, it is evident that the PGD method significantly alters the original image's meaning, indicating an inability to preserve the original content. ProbCW and StAdv perform somewhat better, yet they falter, especially when '0' and '1' are the original digits: for '0', the roundness is compromised; for '1', most adversarial examples take on the form of the target class. Furthermore, ProbCW examples exhibit noticeable overlapping shadows, and the StAdv samples clearly show signs of tampering. In contrast, our method maintains the integrity of the original image's meaning the most effectively.

## 6.2 Imagenet

**Setting**  We employ a diffusion model that has been fine-tuned on $x_{\text{ori}}$ to approximate the distance distribution $p_{\text{dis}}(\cdot|x_{\text{ori}})$. Specifically, we start with a pre-trained diffusion model $p_\theta(x_{t-1}|x_t)$, and then we fine-tune it on a given $x_{\text{ori}}$, as introduced in section 4.1. For the victim distribution, we choose ResNet50 as the surrogate classifier and utilize $f_{\text{CE}}$, the cross-entropy function for $f$.

To evaluate our method's performance, we compare it with several existing approaches: ACE [54], ColorFool [35], cAdv [2] and NCF [51]. Our method is evaluated across three hyperparameter

configurations: $c = 5$, $c = 10$, and $c = 20$. We test the transferability of these methods on Inception V3 [40], EfficientNet B7 [41], VGG19 [36], Resnet 152 [15] and DenseNet 161 [21]. We also list the attack success rate on the adversarial defence methods such as adversarially trained Inception V3 [23], adversarially trained EfficientNet [47] and ensemble adversarial Inception ResNet v2 [42].

**Quantitative Results** We randomly select 1,000 non-human images from the ImageNet dataset to serve as original images $x_{\text{ori}}$, adhering to the ethical guidelines outlined in Section 8.4. For each method, we then generate one untargeted adversarial example per original image. As with the MNIST experiment, Table 2 presents the quantitative results for the ImageNet dataset. In this context, human annotators were presented with pairs consisting of an original image and its corresponding adversarial example and were asked to identify the computer-modified photo. A case is considered successful if the annotator mistakenly identifies the original image as the manipulated one. Therefore, a 'Human Anno.'

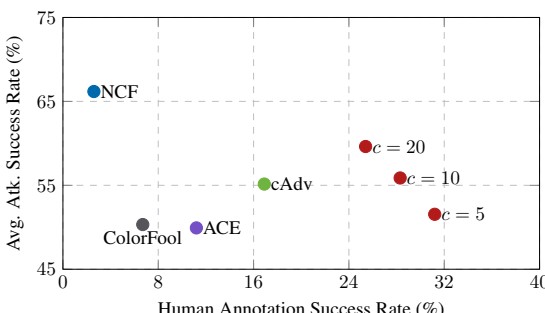

Figure 5: Average attack success rate across the blackbox transferability and defence methods v.s. human annotation success rate illustrated in Table 2.

success rate around 50% suggests that the adversarial examples are indistinguishable from the original images by human observers.

The data in Table 2 places our proposed method second in terms of transferability across different classifiers and defense methods. Note that the 'Average' line is the average of the transferability lines and the adversarial defence lines. Drawing on data from Table 2, Figure 5 is plotted, illustrating that our proposed method not only secures a relatively high attack success rate but also remains minimally detectable to human observers. It's important to mention that while NCF achieves the highest attack success rate in many instances, it is also easily detectable by humans. This observation is supported by the human annotation success rate and further evidenced by our qualitative comparison in Figure 4.

**Qualitative Result** Figure 4 displays adversarial examples generated by our method compared with those from alternative methods, under the same parameters used in the quantitative analysis. The images reveal that other methods tend to produce significant color changes to the original image, rendering the alterations easily recognizable by humans. This observation is corroborated by the 'Human Anno.' row in Table 2. Meanwhile, adversarial examples from our method are more subtle and the alterations are less detectable by humans.

# 7 Related Work

The term 'unrestricted adversarial attack' refers to adversarial attacks that are not confined by geometrical constraints. Unlike traditional attacks that focus on minimal perturbations within a strict geometric framework, unrestricted attacks often induce significant changes in geometric distance while preserving the semantics of the original image. These methods encompass attacks based on spatial transformations [46, 1], manipulations within the color space [19, 54], the addition of texture [2], and color transformations guided by segmentation [35, 51]. Notably, [38] introduced a concept also termed 'unrestricted adversarial attack'; however, in their context, 'unrestricted' signifies that the attack is not limited by the presence of an original image but rather by an original class.

Recent works [7, 50, 6, 5] incorporate adversarial attack gradients into the image editing process and utilize contemporary probabilistic generative models - diffusion models - to create semantic-aware adversarial examples. Our approach is distinct. While all methods involve some form of combining attack gradients with generative gradients, our method is principled, derived from the original optimization problem introduced in Equation (1). Moreover, we introduce a novel probabilistic perspective on adversarial attacks for the first time.

# 8   Discussions

## 8.1   Contrasting MNIST and ImageNet Experiments

**Targeted vs. Untargeted Attacks**   The MNIST dataset, comprising only 10 classes, allows us to perform targeted attack experiments efficiently. However, ImageNet, with its extensive set of 1,000 classes, presents practical challenges for conducting targeted attacks on each class individually. Consequently, we assess untargeted attack performance, aligning with methodologies in other studies.

**Data Diversity**   Adversarially trained networks like MadryNet for MNIST are difficult to fool, primarily due to the limited diversity among handwritten digits. As illustrated in Figure 2 (b), the classifier can nearly memorize the contours of each digit, given its impressive generative capabilities for such data. In attacking this classifier, we carefully selected the $f_{\text{CW}}$ method for the victim distribution to reduce the influence of the target class's 'shadow.' In contrast, for ImageNet, the vast diversity and the relatively weaker generative ability of the victim classifier allow for the use of $f_{\text{CE}}$, facilitating higher confidence in target class recognition by the victim classifier.

## 8.2   Defending This Attack

Adversarial training operates on the principle: 'If I know the form of adversarial examples in advance, I can use them for data augmentation during training.' Thus, the success of adversarial training largely depends on foreknowledge of the attack form. Our method bypasses adversarially trained classifiers because the 'semantic-preserving perturbation' we employ is unforeseen by the classifier designers - they use conventional adversarial examples for training.

Conversely, if designers anticipate attacks from our algorithm, they could incorporate examples generated by our method into their training process - essentially, a new form of adversarial training.

This scenario transforms adversarial attacks and defenses into a game of Rock-Paper-Scissors, where anticipating the type of attack becomes crucial. One might consider training a classifier using all known types of attacks. However, expanding the training data too far from the original distribution typically leads to decreased performance on the original classification task, which is undesirable [52]. We believe that investigating the trade-off between this 'generalized' adversarial training and accuracy on the original task represents a promising avenue for future research.

## 8.3   Limitations

Training or fine-tuning a model for each original image $x_{\text{ori}}$ is time-consuming. Recent advancements, such as faster fine-tuning methods [20, 48], offer potential solutions to mitigate this issue. We see promise in these developments and consider their application an avenue for future research.

## 8.4   Ethical Considerations in User Studies

As mentioned by [28], the ImageNet dataset contains elements that may be pornographic or violate personal privacy. To mitigate the exposure of human annotators in our experiments (see Section 6) to such sensitive content, we avoid selecting any images that depict humans for our original images $x_{\text{ori}}$.

# 9   Conclusion

This paper offers a probabilistic perspective on adversarial examples, illustrating a seamless transition from 'geometrically restricted adversarial attacks' to 'unrestricted adversarial attacks.' Building upon this perspective, we introduce two specific implementations for generating adversarial examples using EBMs and diffusion models. Our empirical results demonstrate that these proposed methods yield superior transferability and success rates against adversarial defense mechanisms, while also being minimally detectable by human observers.

## Acknowledgments and Disclosure of Funding

Andi Zhang is supported by a personal grant from Mrs. Yanshu Wu. Mingtian Zhang acknowledges funding from the Cisco Centre of Excellence. We thank the Ethics Committee of the Department of Computer Science and Technology, University of Cambridge, for their guidance and advice regarding the user study conducted in this work.

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

# A Proof of the Theorems

**Theorem 1.** *Given the condition that $p_{vic}(x_{adv}|y_{tar}) \propto \exp(-c_2 f(x_{adv}, y_{tar}))$, $p_{dis}(x_{adv}|x_{ori}) \propto \exp(-c_1 \mathcal{D}(x_{ori}, x_{adv}))$, the samples drawn from $p_{adv}$ will exhibit the same distribution as the adversarial examples derived from applying the box-constrained Langevin Monte Carlo method to the optimization problem delineated in equation (1).*

*Proof.* [24] introduced the Projected Stochastic Gradient Langevin Algorithms (PSGLA) to address box-constraint optimization problems. By leveraging the PSGLA, we can generate samples close to the solution of the optimization problem as stated in Equation (1). This leads us to the following update rule:

$$x_0 \sim p_0, \quad x_{t+1} = \Pi_{[0,1]^n}\left(x_t - \frac{\epsilon^2}{2}\nabla_x \mathcal{L}(x_t, y_{\text{tar}}) + \epsilon z_t\right), \quad z_t \sim \mathcal{N}(0, I) \tag{7}$$

where $\Pi[0,1]^n$ is a projection that clamps values within the interval $[0,1]^n$. According to [24], samples generated via this update rule will converge to a stationary distribution, which can be termed the Gibbs distribution $p_{\text{gibbs}}$:

$$\begin{aligned}
p_{\text{gibbs}}(x_{\text{adv}}|y_{\text{tar}}) &\propto \exp(-\mathcal{L}(x_{\text{adv}}, y_{\text{tar}})) \\
&\propto \exp(-c_1 \mathcal{D}(x_{\text{ori}}, x_{\text{adv}}) - c_2 f(x_{\text{adv}}, y_{\text{tar}})) \\
&\propto \exp(-c_1 \mathcal{D}(x_{\text{ori}}, x_{\text{adv}})) \exp(-c_2 f(x_{\text{adv}}, y_{\text{tar}})) \\
&\propto p_{\text{dis}}(x_{\text{adv}}|x_{\text{ori}}) \, p_{\text{vic}}(x_{\text{adv}}|y_{\text{tar}})
\end{aligned}$$

which matches the form of $p_{\text{adv}}$. It is a well-established fact that random variables with identical unnormalized probability density functions share the same distribution. $\square$

**Theorem 2.** *Let $p_{vic}(x|y_{tar}) \propto \exp(-cf(x, y_{tar}))$ and $p_\theta(x_{t-1}|x_t) = \mathcal{N}(x_{t-1}; \mu_\theta(x_t), \Sigma_\theta(x_t))$, we have*

$$p_{vic}(\hat{x}_{0|t}|y_{tar})^{1/T} p_\theta(x_{t-1}|x_t) \approx \mathcal{N}\left(\mu_\theta(x_t) + \frac{c}{T}\Sigma_\theta(x_t)g, \Sigma_\theta(x_t)\right)$$

*where $g = -\nabla_{x_{t-1}} f(\hat{x}_{0|t}, y_{tar})|_{x_{t-1}=\mu_\theta(x_t)}$ and $\hat{x}_{0|t}$ is defined in Equation (5).*

*Proof.* For brevity, denote $\mu = \mu_\theta(x_t)$ and $\Sigma = \Sigma_\theta(x_t)$. As suggested by [10], with an increasing number of diffusion steps, $\|\Sigma\| \to 0$, allowing us to reasonably assume that $\log p_{\text{vic}}(\hat{x}_{0|t}|y_{\text{tar}})$ has low curvature relative to $\Sigma^{-1}$. We approximate $\log p_{\text{vic}}(\hat{x}_{0|t}|y_{\text{tar}})$ using Taylor expansion around $x_{t-1} = \mu$ (noting $\hat{x}_{0|t}$ as a function of $x_{t-1}$) as follows:

$$\begin{aligned}
\log p_{\text{vic}}(\hat{x}_{0|t}|y_{\text{tar}}) &\approx \log p_{\text{vic}}(\hat{x}_{0|t}|y_{\text{tar}})|_{x_{t-1}=\mu} + (x_{t-1} - \mu)\nabla_{x_{t-1}} \log p_{\text{vic}}(\hat{x}_{0|t}|y_{\text{tar}})|_{x_{t-1}=\mu} \\
&= C_1 + (x_{t-1} - \mu)\nabla_{x_{t-1}}(-cf(\hat{x}_{0|t}, y_{\text{tar}}))|_{x_{t-1}=\mu} \\
&= C_1 + (x_{t-1} - \mu)cg
\end{aligned}$$

where $C_1$ is a constant and $g = -\nabla_{x_{t-1}} f(\hat{x}_{0|t}, y_{\text{tar}})|_{x_{t-1}=\mu_\theta(x_t)}$. Then

$$\begin{aligned}
\log(p_{\text{vic}}(\hat{x}_{0|t}|y_{\text{tar}})^{1/T} p_\theta(x_{t-1}|x_t)) &= \log p_\theta(x_{t-1}|x_t) + \frac{1}{T}\log p_{\text{vic}}(\hat{x}_{0|t}|y_{\text{tar}}) \\
&\approx -\frac{1}{2}(x_{t-1} - \mu)^T \Sigma^{-1}(x_{t-1} - \mu) + \frac{c}{T}(x_{t-1} - \mu)g + C_2 \\
&= -\frac{1}{2}(x_{t-1} - \mu - \frac{c}{T}\Sigma g)^T \Sigma^{-1}(x_{t-1} - \mu - \frac{c}{T}\Sigma g) + \frac{c^2}{2T^2}g^T\Sigma g + C_2 \\
&= -\frac{1}{2}(x_{t-1} - \mu - \frac{c}{T}\Sigma g)^T \Sigma^{-1}(x_{t-1} - \mu - \frac{c}{T}\Sigma g) + C_3
\end{aligned}$$

which is the unnormalized log pdf of a Gaussian distribution with mean $\mu + \frac{c}{T}\Sigma g$ and variance $\Sigma$. $\square$

# B Preliminaries (Continued)

## B.1 Langevin Monte Carlo (LMC)

Langevin Monte Carlo (also known as Langevin dynamics) is an iterative method that could be used to find near-minimal points of a non-convex function $g$ [29, 53, 43, 30]. It involves updating the function as follows:

$$x_0 \sim p_0, \quad x_{t+1} = x_t - \frac{\epsilon^2}{2}\nabla_x g(x_t) + \epsilon z_t, \quad z_t \sim \mathcal{N}(0, I) \tag{8}$$

where $p_0$ could be a uniform distribution. Under certain conditions on the drift coefficient $\nabla_x g$, it has been demonstrated that the distribution of $x_t$ in (8) converges to its stationary distribution [8, 30], also referred to as the Gibbs distribution $p(x) \propto \exp(-g(x))$. This distribution concentrates around the global minimum of $g$ [13, 49, 30]. If we choose $g$ to be $E_\theta$, then the stationary distribution corresponds exactly to the EBM's distribution defined in (2). As a result, we can draw samples from the EBM using LMC.

## B.2 Training / Fine-Tuning EBM

To train an EBM, we aim to minimize the minus expectation of the log-likelihood, represented by

$$\mathcal{L}_{\text{EBM}} = \mathbb{E}_{x \sim p_d}[-\log p_\theta(x)] = \mathbb{E}_{x \sim p_d}[E_\theta(x)] - \log Z_\theta$$

where $p_d$ is the data distribution. The gradient is

$$\nabla_\theta \mathcal{L}_{\text{EBM}} = \mathbb{E}_{x \sim p_d}[\nabla_\theta E_\theta(x)] - \nabla_\theta \log Z_\theta$$
$$= \mathbb{E}_{x \sim p_d}[\nabla_\theta E_\theta(x)] - \mathbb{E}_{x \sim p_\theta}[\nabla_\theta E_\theta(x)]$$

(see [37] for derivation, this method is also called contrastive divergence). The first term of $\nabla_\theta \mathcal{L}_{\text{EBM}}$ can be easily calculated as $p_d$ is the distribution of the training set. For the second term, we can use LMC to sample from $p_\theta$ [17].

Effective training of an energy-based model (EBM) typically requires the use of techniques such as sample buffering and regularization. For more information, refer to the work of [11].

## B.3 Training / Fine-Tuning Diffusion Models

[18] presented a method to train diffusion models by maximizing the variational lower bounds (VLB), which is expressed through the following loss function:

$$L_{\text{vlb}} := L_0 + L_1 + \ldots + L_{T-1} + L_T$$

where

$$L_0 := -\log p_\theta(x_0|x_1), \quad L_{t-1} := D_{KL}(q(x_{t-1}|x_t, x_0)||p_\theta(x_{t-1}|x_t)), \quad L_T := D_{KL}(q(x_T|x_0)||p(x_T))$$

Assuming $\Sigma_\theta = \sigma_t^2 I$, where $\sigma_t = \beta_t$ or $\sigma_t = \tilde{\beta}_t$ (with $\tilde{\beta}_t := \frac{1-\bar{\alpha}_{t-1}}{1-\bar{\alpha}_t}\beta_t$), and using the parametrization

$$\mu_\theta(x_t, t) = \frac{1}{\sqrt{\alpha_t}}\left(x_t - \frac{\beta_t}{\sqrt{1-\bar{\alpha}_t}}\epsilon_\theta(x_t, t)\right)$$

[18] proposed a simpler loss function, $L_{\text{simple}}$,

$$L_{\text{simple}} = E_{t,x_0,\epsilon}\left[\|\epsilon - \epsilon_\theta(x_t, t)\|^2\right]$$

which is shown to be effective in practice. Later, [27] suggested that a trainable $\Sigma_\theta(x_t, t) = \exp(v \log \beta_t + (1-v)\log \tilde{\beta}_t)$ could yield better results. Since $\Sigma_\theta$ is not included in $L_{\text{simple}}$, they introduced a new hybrid loss:

$$L_{\text{hybrid}} = L_{\text{simple}} + \lambda L_{\text{vlb}}$$

In this work, we adopt the improved approach as suggested by [27].

# C Pseudocode for Sampling Methods from the Adversarial Distribution

---
**Algorithm 1** Sampling from $p_{\text{adv}}$ by EBM
---

**Input:** Trained / finetuned EBM $E_\theta$ depends on $x_{\text{ori}}$, the function $f$ corresponding to the victim classifier, target class $y_{\text{tar}}$, total time step $T$, noise level $\epsilon$, parameter $c$.
**Output:** Sample $x$
$x \sim N(0, I)$.
**for** $t = 1$ **to** $T$ **do**
    $z \sim N(0, I)$
    $x = x - \frac{\epsilon^2}{2}\left(c\nabla_x f(x, y_{\text{tar}}) + \nabla_x E_\theta(x)\right) + \epsilon z$
**end for**

---

---
**Algorithm 2** Sampling from $p_{\text{adv}}$ by diffusion model
---

**Input:** Trained / finetuned diffusion model $(\mu_\theta, \Sigma_\theta)$, the function $f$ corresponding to the victim classifier, the original image $x_{\text{ori}}$, target class $y_{\text{tar}}$, total time step $T$, variance schedule $\beta_1, \ldots, \beta_T$ and its associate $\alpha_1, \ldots, \alpha_T, \bar{\alpha}_1, \ldots, \bar{\alpha}_T$, parameter $c$.
**Output:** Sample $x_0$
$x_T \sim N(0, I)$.
**for** $t = T$ **to** $1$ **do**
    $\epsilon_\theta(x_t) = \frac{\sqrt{1-\bar{\alpha}_t}}{\beta_t}(x_t - \sqrt{\alpha_t}\mu_\theta(x_t))$
    $\hat{x}_{0|t} = \frac{1}{\sqrt{\bar{\alpha}_t}}(x_t - \sqrt{1-\bar{\alpha}_t}\,\epsilon_\theta(x_t))$
    $x_{t-1} \sim \mathcal{N}(\mu_\theta(x_t) - \frac{c}{T}\Sigma_\theta(x_t)\nabla_{x_t}f(\hat{x}_{0|t}, y_{\text{tar}}), \Sigma_\theta(x_t))$
**end for**

---

# D   Practical Techniques

This section outlines practical techniques employed in the implementation.

## D.1   Data Augmentation by Thin Plate Splines (TPS) Deformation

Thin-plate-spline (TPS) [3] is a commonly used image deforming method. Given a pair of control points and target points, TPS computes a smooth transformation that maps the control points to the target points, minimizing the bending energy of the transformation. This process results in localized deformations while preserving the overall structure of the image, making TPS a valuable tool for data augmentation.

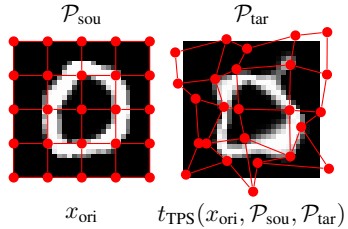

Figure 6: TPS as a data augmentation. **Left**: The original image $x_{\text{ori}}$ superimposed with a $5 \times 5$ grid of source control points $\mathcal{P}_{\text{sou}}$. **Right**: The transformed image overlaid with a grid of target control points $\mathcal{P}_{\text{tar}}$.

As introduced in Section 4, we aim to train an energy-based model on transformations of a single image $x_{\text{ori}}$. In practice, if the diversity of the augmentations of $x_{\text{ori}}$, represented as $t(x_{\text{ori}})$, is insufficient, the training of the probabilistic generative model is prone to overfitting. To address this issue, we use TPS as a data augmentation method to increase the diversity of $t(x_{\text{ori}})$. For each $x_{\text{ori}}$, we set a $5 \times 5$ grid of source control points, $\mathcal{P}_{\text{sou}} = \{(x^{(i)}, y^{(i)})\}_{i=1}^{5 \times 5}$, and defining the target points as $\mathcal{P}_{\text{tar}} = \{(x^{(i)} + \epsilon_x^{(i)}, y^{(i)} + \epsilon_y^{(i)})\}_{i=1}^{5 \times 5}$, where $\epsilon_x^{(i)}, \epsilon_y^{(i)} \sim \mathcal{N}(0, \sigma^2)$ are random noise added to the source control points. We then apply TPS transformation to $x_{\text{ori}}$ with $\mathcal{P}_{\text{sou}}$ and $\mathcal{P}_{\text{tar}}$ as its parameters. This procedure is depicted in Figure 6. By setting an appropriate $\sigma$, we can substantially increase the diversity of the one-image dataset while maintaining its semantic content.

## D.2 Rejection Sampling

Directly sampling from $p_{\text{adv}}(\cdot|x_{\text{ori}}, y_{\text{tar}})$ does not guarantee the generation of samples capable of effectively deceiving the classifier. To overcome this issue, we adopt rejection sampling [44], which eliminates unsuccessful samples.

## D.3 Sample Refinement

After rejection sampling, the samples are confirmed to successfully deceive the classifier. However, not all of them possess high visual quality. To automatically obtain $N$ semantically valid samples[2], we first generate $M$ samples from the adversarial distribution. Following rejection sampling, we sort the remaining samples and select the top $\kappa$ percent based on the softmax probability of the original image's class, as determined by an auxiliary classifier. Finally, we choose the top $N$ samples with the lowest energy $E$, meaning they have the highest likelihood according to the energy-based model.

The entire process of rejection sampling and sample refinement is portrayed in Algorithm 3.

---

**Algorithm 3** Rejection Sampling and Sample Refinement

---

**Input:** A trained energy based model $E(\cdot; x_{\text{ori}})$ based on the original image $x_{\text{ori}}$, the victim classifier $g_\phi$, an auxiliary classifier $g_\psi$, number of initial samples $M$, number of final samples $N$, the percentage $\kappa$.
**Output:** $N$ adversarial samples $x$.
$x = \emptyset$
**for** $0 \leq i < M$ **do**
    $x_{\text{adv}} \sim p_{\text{adv}}(\cdot; x_{\text{ori}}, y_{\text{tar}})$                                  {Sample from the adversarial distribution.}
    **if** $\arg\max_y g_\phi(x_{\text{adv}})[y] = y_{\text{tar}}$ **then**
        $x = x \cup \{x_{\text{adv}}\}$
    **end if**
**end for**
Sort $x$ by $\sigma(g_\psi(x_i))[y_{\text{ori}}]$ for $i \in \{1, \ldots, |x|\}$ in descent order
$x = (x_i)_{i=1}^{\lfloor \kappa |x| \rfloor}$                                 {Select the first $\kappa$ percent elements from $x$.}
Sort $x$ by $E(x_i; x_{\text{ori}})$ for $i \in \{1, \ldots, |x|\}$ in ascent order
$x = (x_i)_{i=1}^{N}$                                           {Select the first $N$ elements from $x$.}

---

## D.4 Adjust the Start Point of Diffusion Process

To preserve more of the original content from $x_{\text{ori}}$ while sampling from its adversarial distribution, we can initiate the diffusion process at a time step $T'$ that is earlier than $T$. The details of this approach are outlined in Algorithm 4.

---

[2]In practice, we could select adversarial samples by hand, but we focus on automatic selection here.

---

**Algorithm 4** Sampling from $p_{\text{adv}}$ by diffusion model

---

**Input:** Trained / finetuned diffusion model $(\mu_\theta, \Sigma_\theta)$, the function $f$ corresponding to the victim classifier, the original image $x_{\text{ori}}$, target class $y_{\text{tar}}$, total time step $T$, num of denoising step $T'$, variance schedule $\beta_1, \ldots, \beta_T$ and its associate $\alpha_1, \ldots, \alpha_T, \bar{\alpha}_1, \ldots, \bar{\alpha}_T$, parameter $c$.
**Output:** Sample $x_0$
$x_{T'} \sim N(\sqrt{\bar{\alpha}_{T'}} x_{\text{ori}}, (1 - \bar{\alpha}_{T'})I)$.
**for** $t = T'$ **to** 1 **do**
    $\epsilon_\theta(x_t) = \frac{\sqrt{1 - \bar{\alpha}_t}}{\beta_t}(x_t - \sqrt{\alpha_t}\mu_\theta(x_t))$
    $\hat{x}_{0|t} = \frac{1}{\sqrt{\bar{\alpha}_t}}(x_t - \sqrt{1 - \bar{\alpha}_t}\,\epsilon_\theta(x_t))$
    $x_{t-1} \sim \mathcal{N}(\mu_\theta(x_t) - \frac{c}{T}\Sigma_\theta(x_t)\nabla_{x_t} f(\hat{x}_{0|t}, y_{\text{tar}}), \Sigma_\theta(x_t))$
**end for**

---

# E  Annotator Interface

Figure 7 and Figure 8 display the interfaces used by annotators in the user study, as described in Section 6. Note that in each case, annotators are given 10 seconds to render their judgment.

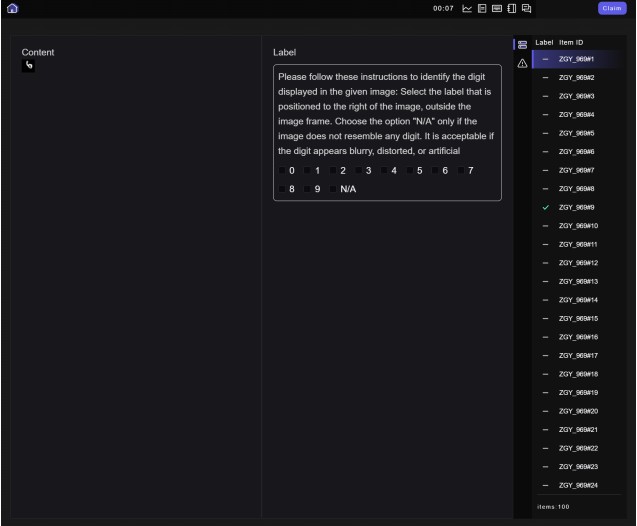

Figure 7: Annotator Interface for image annotation.

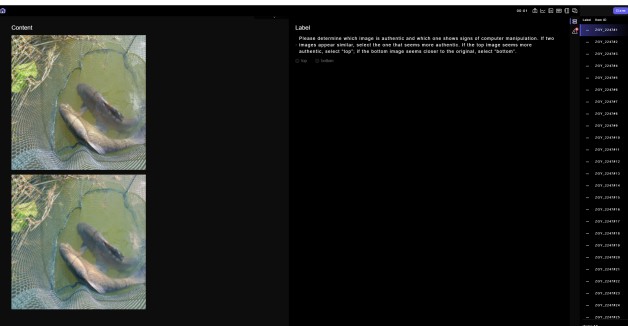

Figure 8: Annotator Interface for comparison.

# F  Broader Impacts of this work

The present study introduces a novel approach: the semantics-aware adversarial attack. This method provides significant insights into the resilience and vulnerability of sophisticated classifiers.

From an advantageous perspective, it highlights the inherent risks associated with robust classifiers. By exposing potential weak points in such systems, the study underscores the necessity for further improvements in classifier security. This can pave the way for building more resilient artificial intelligence systems in the future.

Conversely, the work also presents potential pitfalls. There is a risk that malicious entities might exploit the concepts discussed here for nefarious purposes. It is crucial to take into account the potential misuse of this semantics-aware adversarial attack and accordingly develop preventive measures to deter its utilization for unethical ends.

## G   Datasets and Licenses

We use MNIST [25] and ImageNet [9] in this work. The MNIST dataset is available under the terms of the Creative Commons Attribution-Share Alike 3.0 license. ImageNet does not have a specific license at the moment, but it is a commonly used dataset in the research community.

## H   Compute Resource

We conducted our experiments using multiple workstations, each equipped with an NVIDIA RTX 4090 GPU (24GB VRAM) and 64GB of system memory.

# I Qualitative Result (Continued)

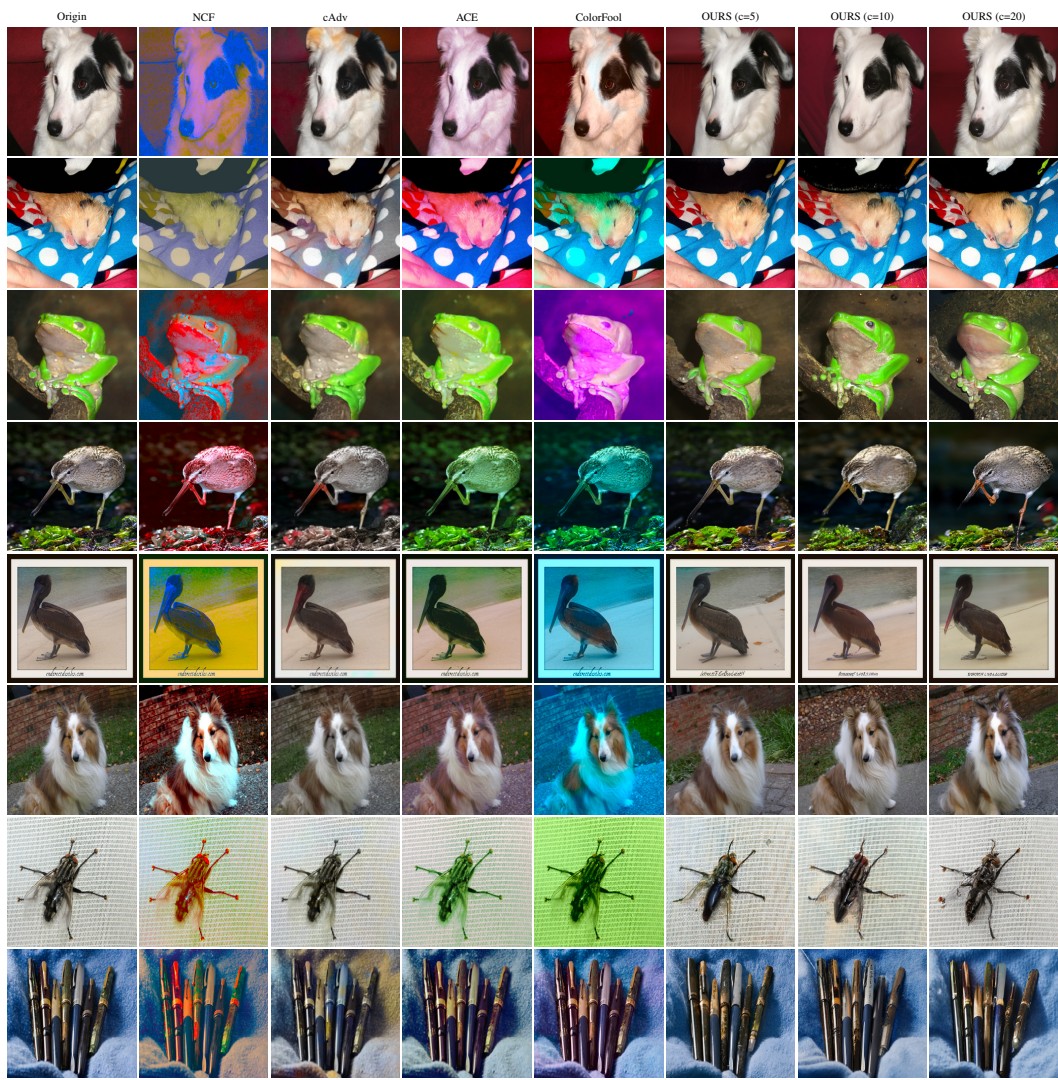

Figure 9: Comparative visual analysis of NCF, cAdv, ACE, ColorFool and our proposed method applied to Imagenet. The surrogate classifier used is ResNet50.

