# OpenReview forum: "Constructing Semantics-Aware Adversarial Examples with a Probabilistic Perspective"
_NeurIPS.cc/2024/Conference — NeurIPS 2024 poster_

### Official Review · Reviewer_aH4C · 2024-07-08

**Soundness:** 2
**Presentation:** 3
**Contribution:** 3
**Rating:** 6
**Confidence:** 4

**Summary:**

This paper tackles the field of adversarial image generation by proposing an unrestricted attack method that can be applied to both targeted and untargeted attacks. The innovative approach considers a probabilistic perspective, treating the victim classifier and geometric constraints as distinct distributions. By drawing adversarial examples from the overlap region, the authors ensure that the semantics of the original image are preserved. The efficacy of this proposed approach is convincingly demonstrated through extensive experiments.

**Strengths:**

1. I find the probabilistic approach proposed in this paper to be particularly innovative and refreshing. The motivation behind this perspective is clearly articulated, providing a solid foundation for the authors' methodology.
2. I am impressed by the encouraging experimental results presented in this paper. The inclusion of a human annotation experiment is particularly noteworthy, as it adds an important layer of validation to the authors' claims. Moreover, the study's success in handling both transfer attacks and adversarial defense scenarios further underscores the model's robustness and effectiveness.

**Weaknesses:**

While the experimental results of the proposed method show promise, I do believe there is room for improvement. Specifically, I think it would be beneficial for the authors to provide more detailed information regarding the human experiment methodology, such as how the five reviewers for the MNIST experiment were selected. Furthermore, I would suggest that the authors consider conducting a follow-up experiment where human annotators are asked to identify perturbed images in the absence of a reference image for the ImageNet experiment, which is a more realistic scenario in an attack setting.

In addition, I find it intriguing that both NCF and cAdv demonstrated higher success rates in generating adversarial examples compared to the proposed method, as shown in Table 2. This highlights some shortcoming in the proposed approach. While it is expected that NCF would generate images that can be identified as perturbed, I am more surprised that cAdv was able to create perturbed images that are highly similar to the original ones.

Lastly, I think it would be beneficial for the authors to explore targeted attacks on ImageNet, given the success of this approach in previous papers such as "Towards Transferable Targeted Attack". This could provide valuable insights into the robustness and effectiveness of the proposed method.

**Questions:**

1. How were the values of c chosen in Table 2?
2. What are the author's thoughts on the tradeoff between choosing different values of c?
3. How were the human annotators selected?
4. How was the set \tau chosen?

---

> ### Author Rebuttal · Authors · 2024-08-06
>
> We appreciate your review and your recognition of our probabilistic perspective and proposed method. Thank you for indicating the oversights and shortcomings in our submission. Your feedback has effectively improved the quality of this work. We respond to your questions and concerns as follows:
>
> ## Response to weaknesses
>
> ### Human Experiments
>
> > more detailed information … such as how the five reviewers for the MNIST experiment were selected.
>
> Regarding the selection of human annotators, please refer to our response to Question 3 below. We have included screenshots of the user interfaces for annotators in Appendix E. We apologize for omitting an important detail in our original submission: we use a voting system to determine the final choice from human annotators. In cases of a draw (which can occur in the MNIST dataset), we randomly select from the tied options. We will incorporate this description in the revised version. Thank you for bringing this to our attention.
>
> > … conducting a follow-up experiment …
>
> We appreciate your suggestion for a follow-up experiment on ImageNet without reference images. Given the time required for human experiments, we will add this experiment in the final version of our paper.
>
> We understand your concern regarding the presence of reference images in our experiment. We would like to clarify that this does not compromise our methodology. In our study, annotators are not informed which image is the reference. We realize we inadvertently omitted a crucial detail in Figure 8: the original image and adversarial example are presented in random order, adhering to the A/B test methodology employed in unrestricted adversarial attack papers such as [1] and [2]. We will emphasize this important point in the final version of our paper.
>
> ### Comparison with NCF and cAdv
> We indeed overlooked emphasizing that our method is optimal in balancing the trade-off between human annotation success rate and attack success rate in the paragraph from lines 258 to 264. In the final version, we will explicitly highlight that, according to Figure 5, our method performs best in this trade-off when appropriate parameters are selected.
>
> We hypothesize that cAdv's lower human annotation success rate may be attributed to the presence of unnatural color spots in the generated images, as illustrated in Figure 4. These artifacts likely make the adversarial examples more noticeable to human annotators.
>
> ### Targeted Attacks on ImageNet
> We acknowledge that targeted attacks on ImageNet pose greater challenges due to the large number of classes, particularly regarding transferability. Our method does not offer specific advantages for this task. In the related work section, we will expand on this issue, discuss the work you recommended, and suggest it as a potential direction for future research based on our probabilistic perspective.
>
> ## Answer to questions
>
> **Q1: How were the values of c chosen in Table 2?**
>
> **Answer:** The parameter c controls the influence of the victim distribution. A larger c results in adversarial samples that are more likely to deceive the victim classifier but may deviate more from the original image's semantics. We initially tested the algorithm on a few images, visually assessing the results. Based on these observations, we selected c values of 5, 10, and 20 as they produced distinct yet reasonable outcomes. We then applied these hyperparameters to attack 1000 ImageNet images, yielding the results shown in Table 2 and Figure 5.
>
> **Q2: What are the author's thoughts on the tradeoff between choosing different values of c?**
>
> **Answer:** Figure 5 illustrates the trade-off between different c values, with the x-axis representing human annotation success rate and the y-axis showing attack success rate. As c increases, adversarial examples become more effective at deceiving the victim classifier but are also more easily identified as adversarial by human annotators. We leave the choice of c to the users of our proposed method, depending on their specific requirements.
>
> **Q3: How were the human annotators selected?**
>
> **Answer:** We conducted human participation experiments through a reputable crowdsourcing company. The five annotators were recruited by this company. Our human experiments passed our institute's ethical review of research, taking into account the crowdsourcing company's qualifications, fair compensation, reasonable workload, and potential risks to workers. Due to the anonymous review process, we cannot share specific details here, but we can provide this information to the Area Chairs if required.
>
> **Q4: How was the set \tau chosen?**
>
> **Answer:** As introduced in Section 4.1, $\mathcal{T}$ represents a set of transformations that we subjectively believe maintain the semantics of the original image. For MNIST, this includes scaling, rotation, and thin-plate spline (TPS) transformations. For ImageNet, we utilize the distribution of the diffusion model after fine-tuning on the original image, which incorporates some implicit natural transformations learned by the model.
>
> If we subjectively believe that appropriate color transformations do not affect semantics, we could incorporate such color transformations into $\mathcal{T}$ when fine-tuning the diffusion model. The primary aim of this work is to propose a probabilistic framework that can embed subjective understanding, rather than to conduct an in-depth exploration of $\mathcal{T}$ selection. We leave this exploratory task for future work.
>
> **Reference**
>
> [1] Song, Yang, et al. "Constructing unrestricted adversarial examples with generative models." Advances in neural information processing systems 31 (2018).
>
> [2] Bhattad, Anand, et al. "Unrestricted adversarial examples via semantic manipulation." arXiv preprint arXiv:1904.06347 (2019).

---

> > ### Comment · Reviewer_aH4C · 2024-08-11
> > **Response to author rebuttal**
> >
> > I thank the authors for responding to my comments and concerns. Some of the questions raised by me have been answered, however not all. Furthermore, I see that other reviewers have raised some valid concerns as well. Thus I have decided not to update my scores.

---

> > > ### Author Response · Authors · 2024-08-14
> > >
> > > Thank you for your feedback. We are dedicated to continuously improving the quality of this paper.

---

### Official Review · Reviewer_ggd1 · 2024-07-12

**Soundness:** 3
**Presentation:** 3
**Contribution:** 3
**Rating:** 7
**Confidence:** 4

**Summary:**

This paper proposes a new type of adversarial attack, which generates adversarial examples by solving a box-constrained non-convex optimization problem. Different from the traditional norm-bounded attacks, this paper focuses on unrestricted adversarial attacks by replacing the geometrical distance measure with a semantically-aware distance measure. Specifically, The authors propose using a Langevin Monte Carlo (LMC) technique to sample adversarial examples from a probabilistic distribution. To preserve semantics, the authors use a learned energy function to guide the generation of adversarial samples. Following this, rejection sampling and refinement techniques are employed to select and improve the quality of the generated samples. Experiments show that this attack can fool classifiers while preserving the semantic information compared to baseline methods.

**Strengths:**

1. This paper introduces an interesting perspective on generating adversarial examples, which is significantly different from the traditional norm-bounded adversarial attacks.
2. This paper is theoretically sound and the proposed solution is very intuitive.
3. It is suprising that the proposed attack can achieve a 100% success rate on an adversarially trained model. Adversarial training is often regarded as a SOTA defense method. Therefore, in my view, this work can motivate researchers in this area to design better defense methods.
4. The proposed method can either outperform baseline methods by a notable margin or significantly improve the quality of the generated adversarial examples in terms of preserving semantic meanings.

**Weaknesses:**

1. Selecting 20 images from each class in the MNIST test set seems to be too little. I understand that it might be infeasible for human annotators to annotate all adversarial images for the entire MNIST, so I would encourage authors to report the success rate except for human annotations using the entire MNIST. I believe this will make the results more convincing.
2. This paper is missing ablation studies for rejection sampling and sample refinement techniques. Is it necessary to include these techniques? How would it affect the attack success rate if one of them is removed?
3. This paper proposes a new attack method but lacks a discussion on how to defend against it. Although it is not compulsory, I am more willing to see how to defend this attack. Can you provide some intuitions on it?
4. Standard deviations are not reported in this paper. Repeated experiments are encouraged.

**Questions:**

Please refer to the **Weaknesses**.

**Limitations:**

Yes.

---

> ### Author Rebuttal · Authors · 2024-08-07
>
> We appreciate the reviewer's thorough evaluation and acknowledgment of our probabilistic approach and proposed methodology. We are grateful for the identification of weaknesses in our submission. The reviewer's insights have significantly enhanced the quality of this work. We address the review points as follows:
>
> **Success rate except for human annotations**
>
> In response to the reviewer's suggestion and to enhance the reliability of our results, we conducted an additional experiment using the MNIST test set, without human annotation. The results are as follows:
>
> |                     | PGD  | ProbCW | stAdv | OURS  | OURS (no tech.) | OURS (rej. samp. only) |
> |---------------------|------|--------|-------|-------|-----------------|------------------------|
> | Human Anno.         | N/A  | N/A    | N/A   | N/A   | N/A             | N/A                    |
> | **White-box**       |      |        |       |       |                 |                        |
> | MadryNet Adv        | 26.9 | 30.5   | 30.0  | 100.0 | 35.4            | 100.0                  |
> | **Transferability** |      |        |       |       |                 |                        |
> | MadryNet noAdv      | 15.4 | 18.0   | 15.8  | 60.4  | 17.9            | 60.2                   |
> | Resnet noAdv        | 9.8  | 10.0   | 11.9  | 23.2  | 13.3            | 21.2                   |
> | **Adv. Defence**    |      |        |       |       |                 |                        |
> | Resnet Adv          | 7.3  | 8.5    | 11.2  | 19.4  | 12.9            | 19.4                   |
> | Certified Def       | 11.2 | 12.1   | 22.3  | 40.6  | 24.7            | 40.9                   |
>
> Note that 'OURS (no tech.)' and 'OURS (rej. samp. only)' refers to the ablation study discussed in the subsequent section.
>
> **Ablation study on two techniques**
>
> We have included the ablation study results for the two techniques in the table above.
>
> Rejection sampling, a classical method, aligns naturally with our probabilistic approach to adversarial attacks. By employing rejection sampling, we can generate adversarial examples that achieve a 100% attack success rate in white-box settings. This perfect success rate is possible because we can consistently reject samples that fail to deceive the victim classifier.
>
> On the other hand, sample refinement primarily affects human annotation while having minimal impact on the attack success rate for classifiers.
>
> These two techniques are only used in MNIST experiments, because generating targeted adversarial samples for an adversarial trained MNIST classifier is relatively hard: As is shown in Figure 2 (b), the adversarially trained MNIST classifier is already have a strong generation ability, meaning that it already memorize the shape of the each digit.
>
> **Discussion on defending this attack**
>
> We appreciate the reviewer pointing out this problem. In the final version of the paper, we will include the following discussion:
>
> Adversarial training operates on the principle: 'If I know the form of adversarial examples in advance, I can use them for data augmentation during training.' Thus, the success of adversarial training largely depends on foreknowledge of the attack form. Our method bypasses adversarially trained classifiers because the 'semantic-preserving perturbation' we employ is unforeseen by the classifier designers - they use conventional adversarial examples for training.
>
> Conversely, if designers anticipate attacks from our algorithm, they could incorporate examples generated by our method into their training process - essentially, a new form of adversarial training.
>
> This scenario transforms adversarial attacks and defenses into a game of Rock-Paper-Scissors, where anticipating the type of attack becomes crucial. One might consider training a classifier using all known types of attacks. However, expanding the training data too far from the original distribution typically leads to decreased performance on the original classification task, which is undesirable [1]. We believe that investigating the trade-off between this ‘generalized' adversarial training and accuracy on the original task represents a promising avenue for future research.
>
> **Standard deviation**
>
> To calculate the standard deviation in repeated experiments, we need to generate multiple sets of adversarial examples for each method and have them annotated by human annotators. Given the time required for annotation, we commit to providing these results in the final version of the paper.
>
> **Reference**
>
> [1] Zhang, Hongyang, et al. "Theoretically principled trade-off between robustness and accuracy." International conference on machine learning. PMLR, 2019.

---

> > ### Comment · Reviewer_ggd1 · 2024-08-10
> >
> > I would like to thank the authors for their thorough rebuttal. My major concerns have been well-addressed. I am now more confident that this paper should be accepted. I have increased my confidence score from 3 to 4.

---

> > > ### Author Response · Authors · 2024-08-14
> > >
> > > Thank you for the feedback and the increased confidence score. We are pleased that our rebuttal has clarified the major concerns.

---

### Official Review · Reviewer_A1FN · 2024-07-12

**Soundness:** 3
**Presentation:** 3
**Contribution:** 3
**Rating:** 6
**Confidence:** 2

**Summary:**

This paper introduces a probabilistic framework for generating adversarial examples, focusing on maintaining the semantic integrity of the original images while implementing substantial pixel-level modifications. Unlike conventional adversarial techniques that rely heavily on minimal geometric perturbations, this approach integrates a semantic understanding into the adversarial example generation process, leveraging energy-based models and diffusion models. The core innovation lies in embedding the semantic interpretation as a probabilistic distribution, which guides the adversarial example generation. This allows for effective deception of classifiers, including those equipped with adversarial defenses, while preserving the semantic content to an extent that remains imperceptible to human observers. Empirical evaluations demonstrate that the proposed method outperforms existing techniques in terms of both effectiveness against defenses and undetectability by humans, establishing a new paradigm for constructing robust and stealthy adversarial attacks.

**Strengths:**

1. The paper is clear and well-written, effectively highlighting its contributions with accessible explanations of complex ideas.

2. This paper presents a new probabilistic framework for generating adversarial examples that goes beyond traditional norm-bounded methods by integrating semantic distributions. The approach is theoretically robust, with the theoretical analysis providing a solid foundation that supports the model's effectiveness and introduces innovative concepts to the field of adversarial machine learning.

3. The proposed method significantly outperforms baseline methods, particularly in preserving their semantic integrity.

**Weaknesses:**

1. The assumption in Equation 4 lacks a detailed derivation, leaving it unclear whether $x_{\text{ori}}$ and $y_{\text{tar}}$ need to be independent. Providing a clear derivation and clarifying this assumption would enhance the theoretical rigor of the paper.

2. The training process for the diffusion models is sensitive and requires careful parameter tuning. The paper does not provide enough detail on this sensitivity or potential solutions to mitigate training instability, which impacts the robustness and reproducibility of the method.

3. The paper does not report standard deviations in the performance results. Repeating the experiments is recommended to ensure the reliability and consistency of the findings.

**Questions:**

Please address the aforementioned concerns and questions.

**Limitations:**

Paper limitations can be found in the above comments.

---

> ### Author Rebuttal · Authors · 2024-08-07
>
> We appreciate the reviewer's time and valuable feedback, which has significantly contributed to improving our work. We address the identified weaknesses as follows:
>
> ### About Equation (4)
> Our proposed framework assumes that the adversarial distribution is proportional to the product of the distance distribution and the victim distribution. This assumption separates the distance distribution from the victim distribution, reflecting our understanding that the similarity between objects is not directly related to the classifier being attacked. While this perspective might be debated, we believe it is reasonable within the context of our paper.
>
> Theorem 1 demonstrates that this proposed form of adversarial distribution is consistent with conventional adversarial attacks. In Appendix A, we prove Theorem 1 by starting with the conventional adversarial attack setting (Equation (1)) and deriving the form of the product of $p_\text{dis}$ and $p_\text{vic}$. This derivation can be found above line 423 in our paper.
>
> We appreciate the reviewer's comment and will provide more intuition about separating the distance and victim distributions between lines 71 and 73 to enhance readability.
>
> ### About the training process of diffusion models
> Please refer to the `README.md` file in the ImageNet folder of our attached anonymous GitHub repository (link provided in the abstract). Our code is based on OpenAI's guided diffusion repository, a mature and widely used diffusion implementation. We fine-tune the weights provided by the guided diffusion repository rather than training a diffusion model from scratch.
> The hyperparameters for fine-tuning are specified in our repository. We maintain most of the original training hyperparameters, with two exceptions:
>
> 1. Learning rate: 1e-6 (a commonly used rate for fine-tuning)
> 2. Number of fine-tuning steps: 300 (empirically determined)
>
> We found that after 300 fine-tuning steps on the original image, the fine-tuned diffusion model adequately reflects the original image. These hyperparameters are consistently applied across all images in our evaluation, which we believe is appropriate for our study.
>
> We thank the reviewer for pointing this out. Indeed, merely including this information in the accompanying code is insufficient. In the final version of the paper, we will add an appendix that provides a detailed discussion of the aforementioned parameter selection and fine-tuning process.
>
>
> ### About Repeating the experiments
> To calculate the standard deviation in repeated experiments, we need to generate multiple sets of adversarial examples for each method and have them annotated by human annotators. Given the time required for annotation, we commit to providing these results in the final version of the paper.

---

### Author Rebuttal · Authors · 2024-08-07

We sincerely thank all the reviewers for their recognition and encouragement of the probabilistic perspective and related approaches we proposed. We are also very grateful for the valuable suggestions made by the reviewers. Your insightful suggestions have significantly contributed to the refinement and improvement of this work.

---

### Decision · Program_Chairs · 2024-09-25

**Decision:**

Accept (poster)

**Comment:**

This paper proposed a way to extend the pixel-level adversarial examples to construct semantics-aware adversarial examples. Authors did good job on rebuttal and all reviewers provide positive scores for this paper. AC read all rebuttal and reviews. AC think all comments are reasonable and think this paper should be accepted to NeurIPS.